# Virulence Characteristics and Molecular Typing of Carbapenem-Resistant ST15 *Klebsiella pneumoniae* Clinical Isolates, Possessing the K24 Capsular Type

**DOI:** 10.3390/antibiotics12030479

**Published:** 2023-02-28

**Authors:** Marianna Horváth, Tamás Kovács, József Kun, Attila Gyenesei, Ivelina Damjanova, Zoltán Tigyi, György Schneider

**Affiliations:** 1Department of Medical Biology and Central Electron Microscope Laboratory, Medical School, University of Pécs, 7624 Pécs, Hungary; 2Department of Biotechnology, Nanophagetherapy Center, Enviroinvest Corporation, 7632 Pécs, Hungary; 3Bioinformatics Research Group, Genomics and Bioinformatics Core Facility, Szentágothai Research Centre, University of Pécs, 7624 Pécs, Hungary; 4Molecular Pharmacology Group, Department of Pharmacology and Pharmacotherapy, Neuroscience Centre, Medical School, University of Pécs, 7624 Pécs, Hungary; 5Division of Microbiological Reference Laboratories, National Public Health Center, 1097 Budapest, Hungary; 6Department of Medical Microbiology and Immunology, Medical School, University of Pécs, 7624 Pécs, Hungary

**Keywords:** *Klebsiella pneumoniae*, clinical isolates, carbapenem-resistant, biofilm formation, virulence potential, whole-genome sequencing

## Abstract

*Klebsiella pneumoniae* is an opportunistic pathogen that frequently causes nosocomial and community-acquired (CA) infections. Until now, a limited number of studies has been focused on the analyses of changes affecting the virulence attributes. Genotypic and phenotypic methods were used to characterise the 39 clinical *K. pneumoniae* isolates; all belonged to the pan-drug resistant, widespread clone ST 15 and expressed the K24 capsule. PFGE has revealed that the isolates could be divided into three distinct genomic clusters. All isolates possessed *allS* and *uge* genes, known to contribute to the virulence of *K. pneumoniae* and 10.25% of the isolates showed hypermucoviscosity, 94.87% produced type 1 fimbriae, 92.3% produced type 3 fimbriae, and 92.3% were able to produce biofilm. In vivo persistence could be supported by serum resistance 46.15%, enterobactin (94.87%) and aerobactin (5.12%) production and invasion of the INT407 and T24 cell lines. Sequence analysis of the whole genomes of the four representative strains 11/3, 50/1, 53/2 and 53/3 has revealed high sequence homology to the reference *K. pneumoniae* strain HS11286. Our results represent the divergence of virulence attributes among the isolates derived from a common ancestor clone ST 15, in an evolutionary process that occurred both in the hospital and in the community.

## 1. Introduction

*Klebsiella pneumoniae* is one of the most important Gram-negative bacteria causing nosocomial and community-acquired (CA) infections, resulting in pneumonia, urinary tract infections (UTI), septicaemia, bronchitis and intra-abdominal infections [1]. *K. pneumoniae* is one of the most prevalent causes of catheter-associated urinary tract infections (CAUTIs), next to *Escherichia coli*. A high incidence of CAUTIs has substantial costs related to extended periods of hospital access. Furthermore, CAUTIs can affect the kidneys and enter the bloodstream causing systemic disease such as septicaemia [2]. Carbapenem-resistant *Klebsiella pneumoniae* (CR-Kp) has become a significant global public health challenge [3].

Severeness of the infection is basically determined by (i) physiological state of the host and (ii) virulence potential of the pathogenic agent. Hospitalization, antimicrobial therapy, prolonged use of invasive medical devices, and major surgeries are considered as the most important environmental factors that contribute to infection [4]. CR-Kp is a major concern for nosocomial infections worldwide since emerging antibiotic resistance makes treatments difficult and are more frequently associated with high morbidity and mortality rates [5]. The therapeutic challenge is large, because CR-Kp isolates are resistant against not only a wide variety of antibiotics, but also against last resorts of clinically used drugs, like carbapenems and colistin [6,7]. 

Today, the factors considered to be important in the survival and virulence of *K. pneumoniae* are the capsule, lipopolysaccharide (LPS), fimbriae, and siderophores (iron chelators). Certain capsule types together with the LPS layer, protect the *K. pneumoniae* against phagocytosis and the bactericidal activity of serum and enable survival and endurance in and out of the host. The phase variable (*fim* switch) type 1 fimbriae, are found to be frequently present in strains of *K. pneumoniae* [8] and mediate adhesion to mannose-containing structures. Type 3 fimbriae are present in almost all *K. pneumoniae* isolates and mediate binding in vitro to human-derived extracellular matrix (ECM) proteins [2] prior to biofilm formation on these surfaces. Its role is pivotal in urinary tract infections (UTI) [9]. The MrkD protein mediates binding to this substrate, whereas the MrkA peptide constitutes the major fimbrial subunit that is polymerized to form the fimbrial shaft [10]. Bacterial biofilms are frequently observed on the surfaces of tissues [11] and biomaterials [12] at the site of persistent infections. Once in the biofilm, extracellular polymeric substance shield bacteria from opsonisation and phagocytosis [13]. In addition, in vitro experiments have demonstrated that the bacteria in biofilms are considerably less susceptible to antibiotics than their planktonic counterparts [14] and can also ensure survival on abiotic surfaces [15].

*Klebsiella* isolates have been shown to produce high-affinity iron-chelating siderophores, that were shown to contribute to the virulence of *K. pneumoniae* [8]. The role of the catechol-type siderophore, enterobactin, in virulence is still uncertain, in contrast to the widely studied hydroxamate-type siderophore, aerobactin [16]. 

Although dissemination is associated with highly diverse isolates of *K. pneumoniae*, certain clonal groups (CG) are dominant in this process. One group is CG15 in which the pan-drug resistant ST 15 clone belongs [17]. This clone was recently identified worldwide from Brazil [18] to China [19] and Europe [20] and was frequently associated with carbapenemase production [18] and nosocomial infections. Recent epidemiologic surveys have revealed that K24 is the most frequent capsule type linked to ST 15 [18], this capsule type does not belong to the classical virulent capsule types [21], but could contribute to survival and therefore persistence of the bacterium either in the hospital environment or in the community [22,23].

Emerging antibiotic resistance including carbapenem resistant *Klebsiella pneumoniae* (CR-Kp) is one of the most important rising public health threats since the early 2000s [24]. One major concern is the predominance of TEM, SHV and CTX-M type extended-spectrum β-lactamases that mainly mediate resistance against β-lactams. Recently the appearance and emergence of β-lactamases with carbapenem-hydrolysing activity (carbapenemases), like the serine carbapenemases KPC and OXA-48, and the metallo-β-lactamases VIM, IMP and NDM are the most concerning from therapeutic point of view [25,26]. Furthermore CR-Kp isolates have undergone extensive disseminations affecting well characterized regions and countries and certain types of carbapenemases show geographical associations. In CR-Kp isolates the VIM type is the most frequently detected metallo-β-lactamase (MBL) in the Mediterranean region, including Spain and Italy [3,24] and has become endemic in certain countries of this region such as Greece [27]. Molecular analyses of emerged VIM-producing *K. pneumoniae* in the northern European countries revealed genetic relatedness to those of southern and other international complexes [28].

Recently, it became evident that not only the activity of carbapenemase can have an effect on carbapenem resistance, but also the presence of certain mutations in outer membrane proteins. These mutations cause alterations in the amino acid sequences that can lead to protein modification, alter pore size and hydrophobicity and can impede uptake of an antibiotic. Mutations in *ompK*35 and *ompK*36 genes are the most known that are related to carbapenem resistance [29].

According to a recent study, infections caused by KPC-producing strains is more frequently associated with increased mortality compared with other mechanisms of resistance [30].

Recently, several studies have focused on the spread of antibiotic resistance and its evolutionary aspects in and outside of the hospital environment. However, knowledge of *K. pneumoniae* ecology, population structure or pathogenicity is relatively limited [17]. In this study appearance and emergence of CR-KP isolates with the same clonal origins and antibiograms in an English Teaching Hospital offered the possibility to analyse the evolutionary aspects affecting virulence attributes.

## 2. Results

### 2.1. Bacterial Identification and Antimicrobial Susceptibility Testing

Isolates were analysed by MALDI-TOF MS and the results were compared with standard conventional identification. All the isolates (n = 39) were identified at the species level (log (score value) ≥ 2.0). The result of standard biochemical tests showed that, all isolates were negative for indole probe, methyl red test, ornithine- and arginine- decarboxylase test, motility test; and all isolates were positive for the adonite test, Voges–Proskauer test, citrate test, malonate test, urease test, lysine-decarboxylase test, saccharose and lactose test.

All isolates were resistant to penicillins, cephalosporins, carbapenems and fluoroquinolones. Antibiotic resistance profile and MIC values of the *K. pneumoniae* isolates is provided in Appendix A, respectively. The minority of the isolates were susceptible to amikacin (48.71%) and gentamicin (30.76%). All isolates, except one (categorized as intermediate susceptible) were resistant to tobramycin (Appendix A).

Of the 39 *K. pneumoniae* isolates included in the study, 100% (39/39) were shown to produce carbapenemase by the Carbapenem Inactivation Method (CIM).

### 2.2. Chromosomal Macro-Restriction Fragment Polymorphism Analysis with PFGE

Isolates were typed by PFGE of *Xba*I-digested total genomic DNA. Strains were considered to be the same clone if they showed ≥85% pattern similarity, or fewer than six fragment differences in the PFGE profile. Based on macrorestriction profile analysis by PFGE the isolates were grouped into three major clusters (Figure 1).

### 2.3. Presence of Virulence-Associated Genes 

The occurrence of virulence-associated genes was determined with PCR and results are summarized in Table 1. *fimH-1* and *mrkD* genes encoding the type 1 and type 3 fimbrial adhesins, were present in 94.87% and 92.3% of the isolates, respectively. Phenotypic test (Table 2) results were in agreement with the PCR data (Appendix A). The adhesion associated genes, *mrkA*, *mrkJ* and *cf29a* were detected at a prevalence of 92.3%, 92.3% and 10.25%, respectively. The activator of the allantoin regulator gene, *allS* was detected in 100% of isolates. Siderophore genes *entB* (enterobactin) and *iutA* (aerobactin) were detected at a prevalence of 94.87% and 5.13%, respectively. Siderophore phenotypic test results were in agreement with the PCR data. Serum resistance was observed in 46.15% of isolates at 3 h. *traT* gene, involved in resistance to serum, was detected in 46.15% of isolates. *rmpA*, the genetic determinant associated with the HMV phenotype was found in 10.25% of isolates. An enhancer of the colony mucoidity, was detected in 10.25%. *uge* and *wziK24* genes encoding the uridine diphosphate galacturonate 4-epidermase gene and K24 capsular type gene, respectively, were detected in 100% of isolates. *magA* gene encoding mucoviscosity-associated gene A, causes hypermucoviscosity, as defined by positive results of the string test, were detected in 10.25% of isolates.

### 2.4. Virulence Associated Phenotypic Assays 

Presence of certain virulence attributes were revealed with phenotypic tests and results are summarized in Table 2. MSHA specific to type 1 fimbriae was detected in 94.87% of the isolates, while incidence of MRHA specific to type 3 fimbriae was 92.3%. 

The ability to form biofilm was detected in 92.3% of the isolates, of which, 34 isolates were high producers, 2 isolates were medium producers, and 3 isolates were poor biofilm producers (Table 2). All isolates that were able to form biofilm also produced type 3 fimbriae. No relationship was revealed between biofilm formation and isolate origin. 

Phenotypic tests aiming to reveal the capacities of the CR-Kp isolates to produce siderophores showed that 94.87% of them produced enterobactin, while only 5.12% produced aerobactin (Table 2). In two cases (53/1 and C8/15) the presence of both iron scavenging systems were revealed, while in one case (C1/16) none of the tested siderophore systems could be detected.

These isolates generally exhibited poor survival in human serum, with only 46.15% of isolates proving resistant to serum bactericidal activity after 3 h (Table 2). Similarly, HMV was found in only 10.25% of isolates.

Cell internalisation assays performed on two human epithelial cell lines (INT 407 and T24) have revealed that most of the isolates (31/39; 79.48%) were able to invade one or both of the cell lines to some extent during a 3 h co-incubation (Figure 2). Fifteen isolates were internalized by both cell lines, while 16 isolates could only invade T24, but not INT407 to a detectable extent. Nine isolates were not able to invade any of the applied cell lines. No general relationship between the origin of isolation and the ability to interact with epithelial cells was established. 

### 2.5. Genome Sequencing and Bioinformatic Analysis of K. pneumoniae Isolates

A representative member of clinical *K. pneumoniae* strains, from four different isolation sites, were selected for whole-genome sequencing.

*K. pneumoniae* strain 11/3 was isolated from a faecal sample. De novo assembly generated 5,517,254 bp. The average contig length was 134,567 and the average GC content of the chromosome was 57.3%. The genome contained 5180 genes and 81 tRNA genes. Analysis of the strain 11/3 sequence data revealed the presence of three types of plasmids, IncFIB (K), IncFII(K) and ColpVC. Based on ResFinder results, the presence of 21 genes related to antibiotics, including the previously characterized *bla*_CTX-M-15_, *bla*_SHV-106_ and bla_VIM-4_ genes, were identified. Several genes and mutations associated with resistance to antimicrobials were detected. Mutations were identified in the *ompK35* (p.A217S) and *ompK36* (p.I128M, p.I70M) genes that are associated with carbapenem resistance. Seven fluoroquinolone resistance mutations (*acrR* p.F197I, p.K201M, p.L195V, p.G164A, p.R173G, p.F172S and p.P161R) and one tigecycline resistance mutation (*ramR* p.A19V) were also found in the genome. Based on VFDB results, 27 virulence genes were analysed (Appendix A), of which 3 virulence genes (*manB*, *manC* and *wbbM*) were only present in strain 11/3. The nucleotide sequence of strain 11/3 was deposited in the GenBank database under the accession number JAJTNS000000000. The general features of genomic analysis of *K. pneumoniae* strain 11/3 are summarized in Table 3 and are shown in Figure 3. 

*K. pneumoniae* strain 50/1 was isolated from blood culture. Shotgun sequences were assembled into one circular replicon measuring 5,379,211 bp. The average contig length was 298,845 and the average GC content of the chromosome was 57.4%. The genome contained 5022 genes and 73 tRNA genes. Analysis of the strain 50/1 sequence data has revealed the presence of one type of plasmid, IncFII(K). Based on ResFinder results, the presence of 14 genes related to antibiotics, including the previously characterized *bla*_CTX-M-15_ and *bla*_SHV-106_ genes were identified. Mutations were identified in the *ompK35* (p.A217S) and *ompK36* (p.I128M, p.I70M) genes that are associated with carbapenem resistance. Seven fluoroquinolone resistance mutations (*acrR* p.K201M, p.F172S, p.F197I, p.G164A, p.L195V, p.R173G and p.P161R) and one tigecycline resistance mutation (*ramR* p.A19V) were also found in the genome. Based on VFDB results, we analysed 29 virulence genes (Appendix A), of which 3 virulence genes (*mrkB*, *mrkH* and *mrkJ*) were only present in strain 50/1. The nucleotide sequence of strain 50/1 was deposited in the GenBank database under the accession number JAJTNT000000000. The general features of genomic analysis of *K. pneumoniae* strain 50/1 are summarized in Table 3 and are shown in Figure 3. 

*K. pneumoniae* strain 53/2 was isolated from sputum. De novo assembly of the genome generated 5,269,114 bp. The average contig length was 292,728 and the average GC content of the chromosome was 57.4%. The genome contained 4908 genes and 81 tRNA genes. Analysis of the strain 53/2 sequence data revealed the presence of one type of plasmid, ColpVC. Based on ResFinder results, 20 genes related to antibiotics, including the previously characterized *bla*_CTX-M-15_, *bla*_SHV-106_ and bla_VIM-4_ genes were identified. Several genes and mutations associated with resistance to antimicrobials were detected. Mutations were identified in the *ompK35* (p.A217S) and *ompK36* (p.I128M, p.I70M) genes that are associated with carbapenem resistance. Seven fluoroquinolone resistance mutations (*acrR* p.F172S, p.P161R, p.K201M, p.L195V, p.F197I, p.G164A and p.R173G) were found in the genome. Based on VFDB results, we analysed 18 virulence genes (Appendix A). The nucleotide sequence of strain 53/2 was deposited in the GenBank database under the accession number JAJTNR000000000. The general features of genomic analysis of *K. pneumoniae* strain 53/2 are summarized in Table 3 and are shown in Figure 3. 

*K. pneumoniae* strain 53/3 was isolated from urine. De novo assembly generated 5,201,283 bp. The average contig length was 273,751 and the average GC content of the chromosome was 57.4%. The genome contained 4844 genes and 63 tRNA genes. Analysis of the strain 53/3 sequence data revealed the presence of one type of plasmid, IncFII(K). Based on ResFinder results, 21 genes were related to antibiotics, including the previously characterized *bla*_CTX-M-15_ and *bla*_SHV-106_ genes. Several genes and mutations associated with resistance to antimicrobials were detected. Mutations were identified in the *ompK35* (p.A217S) and *ompK36* (p.I128M, p.I70M) genes that are associated with carbapenem resistance. Seven fluoroquinolone resistance mutations (*acrR* p.R173G, p.F197I, p.G164A, p.P161R, p.K201M, p.F172S and p.L195V) and one tigecycline resistance mutation (*ramR* p.A19V) were also found in the genome. Based on VFDB results, we analysed 37 virulence genes (Appendix A), of which 5 virulence genes (*cpsAPC*, *galF*, *irp1/ybt*, *irp2/ybt* and *wzi*) were only in strain 53/3. The nucleotide sequence of strain 53/3 was deposited in the GenBank database under the accession number JACTNU010000000. The general features of genomic analysis of *K. pneumoniae* strain 53/3 are summarized in Table 3 and are shown in Figure 3. 

## 3. Discussion

Evolutionary aspects of antibiotic resistance spread and persistence and its impact on the clinical outcome in and out of the hospital environment has been the focus of recent studies [31,32,33,34,35,36]. It has been concluded that the CR-Kp infection-related mortality rate was higher than those of extended-spectrum β-lactamases (ESBL)-producing, and wild-type susceptible *K. pneumoniae* strains [37,38]. Moreover, infection with carbapenem-resistant strains is a risk factor for infection-related mortality [39,40,41,42,43]. Clones with common clonal origins, first identified in an English teaching hospital in 2010, provided a good opportunity to perform a comparative analysis focusing on virulence attributes.

In this study, the clones used were the first CR-Kps isolated in this hospital and originated from the first wave of the emergence (2010–2017). A common origin of the clones was confirmed as they all belonged to ST 15, possessed the moderately virulent capsule type K24, had the same antibiogram (Appendix A), and showed high sequence homology based on PFGE and partially by whole genome sequence analysis. Well documented controls were used, including the hypervirulent *K. pneumoniae* strain NTUH K-2044 [44] and the moderately virulent reference strain MGH78578 [45]. Many attributes were common, but the slight differences among them indicated a divergence or evolutionary separation via an evolutionary process. 

The common clonal origins of the *K. pneumoniae* isolates were supported by the fact that all isolates (n = 39) belonged to ST15 and possessed the K24 capsular type and their antibiograms showed high similarities to each other (Appendix A). Furthermore, identified point mutations (location inside the gene is labelled with: p.XY) in the resistance genes of the four sequenced isolates 11/3, 50/1, 53/2 and 53/3 were localized at the same positions affecting the carbapenem resistance in the *ompK35* (p.A217S) and *ompK36* (p.I128M, p.I70M) [46]. This was also the case in other two inactive resistance genes, namely the *acrR* and *ramR*. The *acrR* gene (1492–4449 bp) confers resistance to fluoroquinolones [47,48], is located on a plasmid, and was affected with the same seven point mutations (p.F197I, p.K201M, p.L195V, p.G164A, p.R173G, p.F172S and p.P161R). Only one mutation (p.A19V) was identified in all sequenced isolates in the *ramR* gene, that confers resistance to tigecycline [47,48,49].

Expression the type 1 (92.3%) and type 3 (94.87%) fimbriae (Table 2) showed the capacity of the clones to colonize abiotic and biotic surfaces, and this contributed not only to survival but also to pathogenesis. These findings are in agreement with earlier studies which show that the adhesive subunit FimH in particular, of type 1 fimbriae, plays an important role in UTIs caused by *K. pneumoniae* [50]. Type 3 fimbriae assist adhesion to human tissue structures (e.g., lung, kidney) and promote biofilm formation on abiotic surfaces. As such, they may play a role in biofilm-associated infections in catheterized patients [2,51,52,53], but are also present in UTI *E.coli isolates* [54]. Correlation between the presence of *mrkD*, the expression of type 3 fimbriae, and strong biofilm formation was also confirmed by our study (Table 1 and Table 2), indicating its pivotal role in survival and persistence of the isolated clones. Incidence of *mrkD* genes (92.3%) among our CR-Kp isolates correlate with recent findings (94%) [55]. Results of our and this latter recent study are also comparable based on the incidence of *magG* as being 10.25% and 11%, respectively.

The high rate of enterobactin expression (94.87%, Table 2) was consistent with previous studies [56,57,58,59] and further supported the pathogenic potential of the isolates. Furthermore, aerobactin was also reported to be an important determinant of virulence of *K. pneumoniae* [16] if present, although it was more rarely found in *Klebsiella*. Its reported rates range from 3–6% [21], similar to our findings (5.12%). Although in our study no serious outcomes were registered among the patients infected by the isolates, evidence in the literature is mounting that *K. pneumoniae* strains carrying acquired siderophores have enhanced capacities to cause invasive diseases [17,59,60,61,62].

A high rate of siderophore production could also support cell internalization and invasion. The heterogeneous picture of the invasion capacities of the isolates with common clonal origin testify to unknown molecular biological changes that could affect the invasion capacities of these bacteria. Cell internalization in most bacteria is a still an unsolved multifactorial process, that in case of *K. pneumoniae* was first reported more than two decades ago [63]. Because of its shielding effect, internalization is a bacterial survival strategy that could also spoil the efficacy of targeted therapies presenting a challenge to the clinician. Despite these therapeutic consequences, cell internalization is still not in the focus of *K. pneumoniae* research [64,65,66]. In our study, internalization experiments (Figure 2) not only revealed the differences among the isolates, but also revealed that most isolates showed preference for the T24 bladder carcinoma cells, instead of INT 407. 

In contrast, past studies have focused on the HMV phenotype of *K. pneumoniae* and its contribution to hypervirulence [61,67,68]. Our data confirmed the position that HMV and hypervirulence are different phenotypes that should not be described synonymously. This finding is consistent with recent opinions [69,70]. Although HMV positive clones (53/3, 53/11, 50/3, 11/1) can form biofilm, produce enterobactin and resist degradation in human serum, they were isolated either from mild UTIs or from faecal samples. Furthermore, our results confirmed earlier findings that the presence of the *rmpA* gene, encoding a positive regulator of mucopolysaccharides expression, is necessary for the HMV phenotype, since we detected this regulator only in the 4 HMV positive clones. To best of our knowledge this is the first study where the *rmpA* gene has been detected in a CR-Kp strain belonging to the ST 15 clone. 

The described differences among the isolates with common clonal origins could be a result of an evolutionary process happening in the community and subsequently brought into the hospital. Based on the cluster analysis, we hypothesize that members of cluster I may originate from these two epidemiological processes. However, based on the isolation years and genomic similarities between members of cluster II and cluster III, revealed with PFGE, we postulate that two internal hospital epidemics could have occurred in 2011 and 2015, respectively. We have no available community data from the Chester region from this period, but based on the hospital data, the dominance of the ST 15 clone could be speculated to have emerged from 2010–2017. 

Interestingly, sequence analysis revealed that four sequenced clones (11/3, 50/1, 53/2, 53/3) showed the highest genomic similarity to a *K. pneumoniae* strain HS11268 (Figure 3) isolated in China in 2011 [38]. The genomic similarity between *K. pneumoniae* strain 53/3 and HS11286 is 96%, strain 53/2 and HS11286 is 97%, strain 50/1 and HS11286 is 90%, strain 11/3 and HS11286 is 96%, based on NCBI Nucleotide BLAST.

Despite these similarities, it is interesting to note that no serious infections were detected among the patients. One explanation for this is the genetic background of the clones. Most clones (53.85%) were not resistant to the bactericidal effect of human serum. On the other hand, almost all clones possessed the K24 capsule, which in contrast to K1, K2, and K5 [21,71,72] is not considered as an important capsule type in the pathogenic process. Nevertheless, the ability of all clones to form a firm biofilm with the help of the K24 capsule maintains that this capsule type, together with type 3 fimbria, supports the survival of bacteria on abiotic and potentially on biotic surfaces. This survival potential of the K24 capsule type might explain why this is the most frequent capsule type linked to ST 15 [18], an otherwise pan-drug resistant widespread clone recently identified worldwide [18,19,60,72,73,74].

Although we do not know what attributes the original ST 15 clone had, we hypothesize that after its appearance this clone underwent, and is still currently undergoing a homing process, when the bacterium adapts to survival in the hospital environment by losing some of its virulence attributes, such as the ability to survive in serum and to invade eukaryotic cells. Such a process in the hospital, community, or in the environment is an important aspect of bacterial evolution with relevance to human pathogenesis and was also formerly outlined in cases of *S. aureus* and *L. pneumophilia* [75,76]. Till now no, relevant data were found in case of *Klebsiella pneumoniae*. This was the reason that our study mainly focused on the divergence of the virulence attributes among isolates with common clonal origin. In addition, sequence analysis could also provide an insight into the contributing mechanisms of the carbapenem resistance in the isolates i.e., mutations in *ompK35* and *ompK36.* Isolates were resistant to cefoxitin (Appendix A), indicating that the mutations detected by WGS analysis in the *ompK35* and *ompK36* truly manifested.

## 4. Materials and Methods

### 4.1. Bacterial Isolates, Growth Conditions

Thirty-nine *K. pneumoniae* isolates (one isolate per patient) collected from the Microbiology Laboratories of University Hospital in Chester, England were studied (Table 4). They were recovered from different specimens: faecal (n = 21), urine (n = 11), sputum (n = 4), and blood cultures (n = 3). Identification of the isolates was performed by Matrix-Assisted Laser Desorption/Ionization Time-of-Flight Mass Spectrometry (MALDI-TOF MS, Microflex, Bruker Daltonics, Billerica, MA, USA) and was confirmed with the standard biochemical tests (indole, adonite, Voges-Proskauer, methyl red, citrate, malonate, urease, lysine-, ornithine-, arginine- decarboxylase, saccharose, lactose and motility tests) prior to the study. 

Bacteria were routinely grown on 37 °C in Luria–Bertani (LB) broth or on eosin–methylene blue (EMB) agar (Difco, Fisher Scientific, Leicestershire, UK) plates. Special media and control strains for fimbria, biofilm, and siderophore production tests, are indicated at relevant sections. *K. pneumoniae* strains NTUH-K2044 [44] and MGH 78578 [45] were used as controls for each experiment. Other control strains are designated in the relevant assays.

### 4.2. Antimicrobial Susceptibility Testing

Antimicrobial susceptibility was determined with the disc diffusion method on Mueller–Hinton (MH) agar (Oxoid, Basingstoke, UK). The inoculum was adjusted to an optical density of 0.5 McFarland, and the plates were incubated for 18 h at 35 °C. Resistance against the following antibiotics were tested: AMP: Ampicillin 10 µg, AMC: Amoxicillin/Clavulanic acid 20/10 µg, PTZ: Piperacillin/Tazobactam 30/6 µg, SAM: Ampicillin/Sulbactam 10/10 µg, FEP: Cefepime 30 µg, CTX: Cefotaxime 5 µg, CAZ: Ceftazidime 10 µg, CRO: Ceftriaxone 30 µg, CXM: Cefuroxime 30 µg, FOX: Cefoxitin 30 μg, ETP: Ertapenem 10 µg, IMP: Imipenem 10 µg, MEM: Meropenem 10 µg, CIP: Ciprofloxacin 5 µg, LEV: Levofloxacin 5 µg, AN: Amikacin 30 µg, GM: Gentamicin 10 µg and TM: Tobramycin 10 µg. Antimicrobial disks were purchased from Oxoid, Hungary. *E. coli* ATCC 25922 was used as a control. The results were interpreted according to the guidelines of the European Committee on Antimicrobial Susceptibility Testing (EUCAST) [77].

A new phenotypic test, called the Carbapenem Inactivation Method (CIM), was used to detect carbapenemase activity [78]. To perform the CIM, a suspension was made by suspending a full 10 μL inoculation loop of culture, taken from a Muller–Hinton (MH) agar (Oxoid, UK) plate in 400 μL Tryptic Soy Broth (TSB). Subsequently, a susceptibility-testing disk containing 10 μg meropenem (Oxoid, UK) was immersed in the suspension and incubated for 4 h at 35 °C. After incubation, the disk was removed from the suspension using tweezers, placed on a MH agar plate inoculated with a susceptible *E. coli* ATCC 29522 indicator strain and incubated for 18 h at 35 °C. Inoculation of the MH agar plate with the indicator strain was done with suspension of OD_595_ 1.25, streaked in three directions using a sterile cotton swab. If the bacterial isolate produced carbapenemase, the meropenem in the susceptibility disk was inactivated allowing uninhibited growth of the susceptible indicator strain. Disks incubated in suspensions that do not contain carbapenemases yielded a clear inhibition zone. Each isolate was tested two times.

### 4.3. Chromosomal Macro-Restriction Fragment Polymorphism Analysis with Pulsed-Field Gel Electrophoresis (PFGE)

Pulsed-field gel electrophoresis (PFGE) was used to reveal the clonal relationship among the *K. pneumoniae* isolates. Strains were grown overnight (10 h) on LB agar at 37 °C. The ODs were adjusted to 1.3 to 1.4 (~10^8^ CFU/mL) at 540 nm. Genomic DNA was prepared in low-gelling point agarose (BioRad, USA) by a procedure developed at the Centre for Disease Control (CDC) [79]. DNAs were in cube digested with *Xba*I (New England Biolabs, San Diego, CA, USA) restriction endonuclease for 12 h, with 10 units/mL. Separation of the fragments was performed by using the CHEF-DR II system (BioRad, Hercules, CA, USA). DNA was electrophoresed for 24 h at 14 °C in a 1.2% agarose gel (Sigma-Aldrich, St. Louis, MO, USA) at 6 V/cm with a linear gradient pulse time of 54 s. Interpretation of PFGE patterns was based on the criteria of Tenover et al. [80]. After photographing, gels were analysed and interpreted with Fingerprinting II Informatix Software (BioRad, USA). Levels of similarities were calculated with the Dice coefficient, and unweighted pair group method with arithmetic averages (UPGMA) was used for the cluster analysis of the PFGE patterns. Pulsotypes (PTs) were defined at 85% similarity between macrorestriction patterns [20].

### 4.4. Multilocus Sequence Typing (MLST) Analysis

Genomic DNA was isolated from bacteria using a Qiagen DNeasy blood and tissue kit (Qiagen, Valencia, CA, USA). Seven housekeeping genes (*rpoB* F: Vic3: GGCGAAATGGCWGAGAACCA, R: Vic2: GAGTCTTCGAAGTTGTAACC*; gapA* F: gapA173: TGAAATATGACTCCACTCACGG, R: gapA181: CTTCAGAAGCGGCTTTGATGGCTT; *mdh F: mdh130:* CCCAACTCGCTTCAGGTTCAG, R: mdh867: CCGTTTTTCCCCAGCAGCAG*; pgi F: pgi1F:* GAGAAAAACCTGCCTGTACTGCTGGC, R: pgi1R: CGCGCCACGCTTTATAGCGGTTAAT; *phoE F: phoE604.1:* ACCTACCGCAACACCGACTTCTTCGG, *R: phoE604.2* TGATCAGAACTGGTAGGTGAT; *infB F: infB1F:* CTCGCTGCTGGACTATATTCG, R: infB1R: CGCTTTCAGCTCAAGAACTTC and *tonB F: tonB1F:* CTTTATACCTCGGTACATCAGGTT, R: tonB2R: ATTCGCCGGCTGRGCRGAGAG) were amplified by polymerase chain reaction (PCR), using conditions and primers designated by the PubMLST *Klebsiella pneumoniae* MLST Database [81], with 50 ng of genomic DNA as a template. The PCR products were purified using a QIAquick PCR purification kit (Qiagen, Valencia, CA, USA). Sequencing was performed at the Countess of Chester Hospital, Institute of Microbiology, Chester, Cheshire, UK. Sequence alignment was performed using MacVector editing software, and the resulting contig was compared against the PubMLST database to determine the designated allele. To identify the sequence type, allelic profiles were generated for each clinical isolate and compared to the MLST database.

### 4.5. PCR Detection of Virulence-Associated Genes

The presence of virulence-associated genes, including the activator of the allantoin regulator gene (*allS*), regulator of mucoid phenotype A (*rmpA*), siderophore genes: enterobactin (*entB*) and aerobactin (*iutA*), adhesion associated genes (*cf29a*, *mrkA*, *mrkJ*, *mrkD*, *fimH-1*), uridine diphosphate galacturonate 4-epimerase gene (*uge*), serum resistance associated gene (*traT*) and mucoviscosity-associated gene A (*magA*) was studied by PCR using primers and conditions described elsewhere (Appendix A).

Genetic determinants of the K24 capsule type were revealed after whole-genome sequencing of four representative strains and analysis of these sequences. Based on the *wzi* cluster analysis, the K24 specific primer pair (Fw: 5′-AGATAATAGG CAACAGCGTTCT-3′ and Rev: 5′-GATACGTTAAA CGCCTCAAGTA-3′) was designed by Primer-BLAST software [82].

The sequenced *K. pneumoniae* strains NTUH-K2044 [44] and MGH 78578 [45] were used as controls.

For DNA extraction, 1.5 mL of overnight (10 h) broth culture was centrifuged at 12,000 rpm for 2 min. The pellet was resuspended in 100 µL sterile deionized water and boiled for 10 min. After a final centrifugation at 12,000 rpm for 10 min, the supernatant containing template DNA was recovered, and used for the analysis. Conditions of the amplification reactions were as follows: 95 °C for 2 min for initial denaturation, which was followed by 34 cycles consisting of 95 °C denaturation for 30 s, annealing temperatures for 30 s, and a 72 °C elongation for 1 min. A one step termination was carried out at 72 °C for 10 min. Samples were electrophoresed in 1% agarose (Invitrogen, Waltham, MA, USA) gels, stained with ethidium bromide (Acros Organics, Antwerp, Belgium), and visualized under UV light.

### 4.6. Phenotypic Tests

#### 4.6.1. Type 1 Fimbriae Assay (Mannose Sensitive Haemagglutination, MSHA)

The traditional *S. cerevisiae* agglutination test was used to detect the expression of the type 1 fimbriae—Mannose Sensitive Haemagglutinins (MSHA)—on the surface of the *K. pneumoniae* isolates. A cell suspension (37 µL; 1 × 10^9^ CFU/mL) of the standard *Sacharomyces cerevisiae* W303 was mixed with the bacterial dilution (1 × 10^8^ CFU/mL) on a glass slide, and gently rotated. To confirm the specificity, tests were also performed in the presence of α-methyl-d-mannoside (5%) (Fluka, Buchs, Switzerland) as it inhibited the specific agglutination [83]. All agglutination assays were carried out three times. *K. pneumoniae* strain 71 and *K. pneumoniae* strain 39 were used as positive controls, and *K. pneumoniae* strain 130 and *K. pneumoniae* strain 131 were used as negative controls.

#### 4.6.2. Type 3 Fimbriae Assay (Mannose-Resistant Haemagglutination, MRHA)

Presence of the type 3 fimbriae mediated mannose-resistant haemagglutination on *K. pneumoniae* isolates was revealed by using the classical method of Podschun and Sahly [84]. Tannic acid (Fluka, Hungary) treated bovine erythrocyte suspension (Culex, Budapest, Hungary) was used. Bacteria were grown at 37 °C on brain heart infusion (BHI) agar plates (Oxoid, UK) for 24 h. Cells were collected, resuspended in physiological saline and their cell counts were adjusted to ~10^8^ cells/mL in physiological saline. Thirty-two µL of bacterial suspensions and 38 µL of erythrocytes were mixed on glass slide, manually rotated, and observed for 10 min at room temperature (22 °C). Agglutination was finally read after further incubation for 5 min at 4 °C [39]. All agglutination assays were carried out three times. *K. pneumoniae* strain 130 and *K. pneumoniae* strain 131 were used as positive controls, and *K. pneumoniae* strain 71 and *K. pneumoniae* strain 79 were used as negative controls.

#### 4.6.3. Biofilm Assay

The biofilm forming capacities of the *K. pneumoniae* isolates were tested by the slightly modified crystal violet binding plate assay [85]. Bacteria were subcultured in LB broth for 12 h at 37 °C, in a shaking incubator (120 RPM). The ODs were adjusted to 0.9 to 1.0 (~10^8^ CFU/Ml) at 540 nm. Twenty µL of the adjusted bacterial cultures and 180 µL LB broth were transferred to 96-well polystyrene microtiter plates (Sarstedt, Nümbrecht, Germany) and incubated for 24 h at 37 °C. Planktonic bacteria were removed with gentle washing and fixed with 2% formalin-phosphate buffered saline (PBS; Sigma-Aldrich, USA). Intensities of biofilm formations were revealed by 1% crystal violet (Sigma-Aldrich, USA) staining for 20 min at room temperature (22 °C), and by solubilizing the layer with 1% SDS (Sigma-Aldrich, USA). Extinction of the solubilized crystal violet in each well was measured at 595 nm with a FLUOstar Optima Microplate Reader (BMG Labtech, Ortenberg, Germany). Controls were performed with crystal violet binding to the wells exposed only to the culture medium without bacteria. Biofilm assays were repeated three times in three independent experiments and in each assay, quantification was performed in four separate wells. *K. pneumoniae* strain 703+ was used as a positive control and *K. pneumoniae* strain 446- was used as a negative control. Isolates were classified as high biofilm-producers (OD ˃ 3.0), medium producers (OD 1.0–3.0) or poor producers (OD < 1.0).

#### 4.6.4. Siderophores Production Assay

Production of enterobactin and aerobactin were detected by the cross-feeding bioassay described by Hantke [86]. Briefly, nutrient agar supplemented with 2,2′-dipyridyl (Sigma-Aldrich, USA) (final concentration 275 µM) was used as iron-restricted agar medium. Growth capacity of the CR-Kp isolates was tested in parallel in the presence of two different indicator strains. *E. coli* H1887 (ColV^−^, Aer^−^, Iut^+^, FepA^−^, Fiu^−^, Cir^−^, aroB) was used as the indicator strain for aerobactin production and strain H1939 (FepA^+^, Fiu^−^, Cir^−^, FhuA^−^, FhuB^−^, aroB) for enterobactin. Aerobactin production was counterchecked with *E. coli* strain H1886, which is the Iut^−^ parent strain of H1887. Strain K311 (pColV-K311) served as a positive control in the aerobactin test [56]. Each isolate was tested three times.

#### 4.6.5. Serum Bactericidal Assay

Normal human serum (NHS), pooled from healthy volunteers, was divided into equal volumes and stored at −20 °C before use. The serum bactericidal activity was measured using the method described by Podschun et al. [56], with slight modification. Bacteria were grown at 37 °C in LB broth for 24 h, with shaking at 100 rpm in an incubator. After washing with physiological saline solution, OD_600_ was adjusted to 0.4. Bacteria were diluted to 2–3 × 10^6^ cell/mL in physiological saline. Then, 25 µL from the bacterial suspensions and 75 µL from the undiluted NHS were mixed in the 96 well polystyrene microtiter plates (Sarstedt, Germany), and incubated at 37 °C. Samples were taken immediately after mixing and after incubation for 1 and 3 h, and serial dilutions were plated on LB agar for colony forming unit (CFU) determination. Resistance was graded by the mean 3 h survival ratio (ratio of colony count after serum treatment for 3 h compared with baseline). The highly serum resistant *K. pneumoniae* strain NTUH-K2044 was used as a positive control [44]. Each strain was tested three times.

#### 4.6.6. Hypermucoviscosity (HMV) Testing

Single colonies obtained after overnight culture on blood agar plates were tested for their ability to form viscous strings when a standard inoculation loop was touched onto their surface and slowly raised. The formation of string greater than 5 mm in length is indicative of the hypermucoviscosity (HMV) positive phenotype [87]. *K. pneumoniae* strain NTUH-K2044 [44] was used as a positive control. Each isolate was tested twice.

### 4.7. Whole-Genome Sequencing

In order to get a detailed view of the genome organization of the ST 15 *K. pneumoniae* lineage, sequencing of four representative members from different specimens was carried out: *K. pneumoniae* strain 11/3 (isolated from faeces), strain 50/1 (isolated from blood culture), strain 53/2 (isolated from sputum) and strain 53/3 (isolated from urine).

DNA was extracted from *K. pneumoniae* strains 11/3, 50/1, 53/2 and 53/3. DNA for whole-genome sequencing were extracted from cultures grown overnight (10 h) in LB agar, using DNA extraction kit (PureLink Genomic DNA Mini Kit, Thermo Fischer Scientific, Waltham, MA, USA) following the manufacturer’s protocol. At the end of the extraction process, DNA samples were dissolved in 100 μL of sterile nuclease free water. Genomic DNA sequencing libraries were prepared using the Nextera XT Library Preparation kit (Illumina, San Diego, CA, USA). Sequencing was performed using MiSeq Reagent Kit v3 (600 cycles) on an Illumina MiSeq (Illumina, San Diego, CA, USA). Assembly of the pure sequence was performed with the MyPro pipeline [88]. Open reding frames were predicted and annotated with the Rapid Annotation using Subsystem Technology [89]. Homology searches were conducted with the BLAST tools available at the NCBI website [90].

Analysis of antibiotic resistance was performed with ResFinder 4.1 [91]. Additional software, including Pathogen Wach [92], was used for analysis of specific plasmid genetic features. Nucleotide sequences for validated *Klebsiella* genus virulence genes were downloaded from the Virulence Factor Database (VFDB) [93]. Nucleotide–nucleotide alignment was run using BLASTN v2.11.0 on macOS 11.6 for our *Klebsiella* sp. contig sequences against the downloaded virulence factor database. Our annotated sequences along with three *Klebsiella pneumoniae* subsp. *pneumoniae* reference genomes (strain HS11286, SAMN02602959; MGH78578, SAMN02603941; NTUH-K2044, SAMD00060934) were visualized using the CGView Server Beta web tool [94]. BLAST alignments for all sequences were run against the nucleotide sequence of *K. pneumoniae* strain 53/3 and visualized. Virulence genes for strain 11/3, 50/1, 53/2 and 53/3 identified previously using the VFDB were marked on the CGView map. 

### 4.8. Cell Internalization Assay

Two human cell lines, the human intestinal cell line INT 407 and the bladder carcinoma cell line T24 were used to reveal the abilities of the investigated CR-Kp isolates to invade cultured cells. Invasion assays were performed essentially as described previously [95]. Briefly, semiconfluent cell monolayers were prepared (3 × 10^5^ cells/well) in RPMI 1640 medium (Lonza, Verviers, Belgium) supplemented with 10% heat-inactivated (30 min for 56 °C) calf bovine serum (Sigma-Aldrich, USA), 10,000 U/mL of penicillin, 10 μg/mL of streptomycin and 0.5 mg/mL of neomycin and incubated overnight (10 h) at 37 °C in a humidified 5% CO_2_ incubator. On the following day, cells were washed with PBS (pH 7) and to each well 1 mL RPMI 1640 medium (Lonza, Belgium) and bacterial suspensions (OD_600_ = 1, ~1 × 10^8^ CFU/mL) were added to reach 10× dilution [96]. Plates were incubated at 37 °C in a humidified, 5% CO_2_ incubator for 3 h. The plates were then washed three times with PBS (pH 7) to remove unbound bacteria. Fresh cell culture medium containing 100 µg/mL Polymixin B was then added to kill all extracellular bacteria and incubated for 1 h at 37 °C. Then wells were washed three times with PBS (pH 7) and lysed with 0.05% Triton X-100 (Sigma-Aldrich, Budapest, Hungary). The intracellularly surviving *K. pneumoniae* cell counts were determined by out-plating. All assays were performed in triplicate and were repeated independently twice. *Salmonella enterica* serotype *Typhimurium* strain ATCC14028, *K. pneumoniae* strain 3091 (accession number: SAMEA8948279) [97] and *K. pneumoniae* strain NTUH-K2044 (accession number: SAMD00060934) were used as positive controls and *K. pneumoniae* strain MGH78578 (accession number: SAMN02603941) was used as a negative control.

## 5. Conclusions

In this study an evolutionary process could be outlined for a CR-Kp ST 15 clone that first appeared in an English Teaching Hospital in 2010. We conclude that during this process, which likely occurred in parallel both in the hospital and the community, divergence of virulence attributes could be observed that support persistence of the original clone rather than virulence. Long term studies in hospital environments, supplemented with community data, could reveal, in the future, the changes in the virulence potentials of emerging clones, similar to changes that are studied for antibiotic resistance. To the best of our knowledge this is the first study where the *rmpA* gene has been detected in a CR-Kp strain belonging to the ST 15 clone.

## Figures and Tables

**Figure 1 antibiotics-12-00479-f001:**
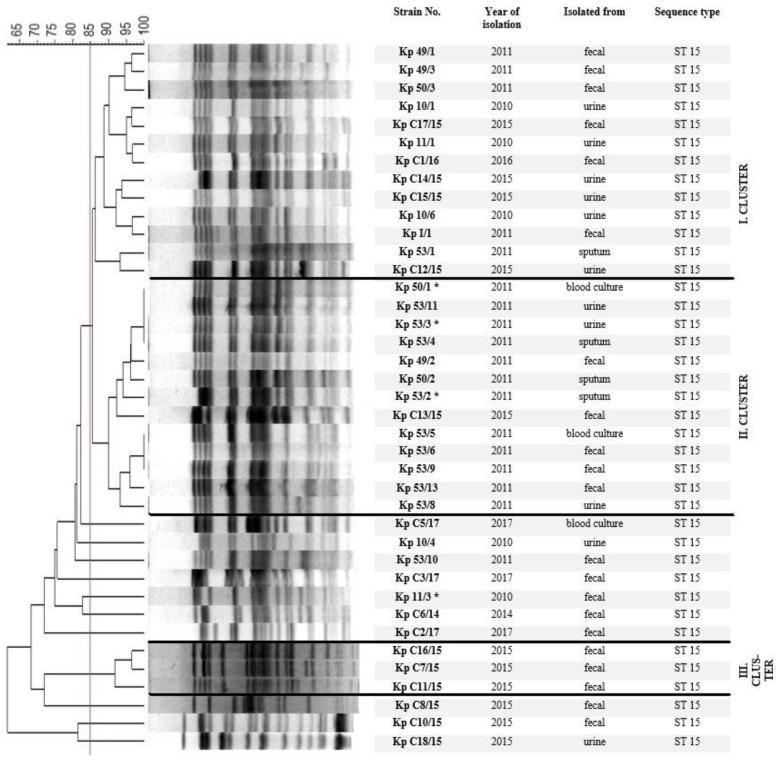
Relationships based on the PFGE profiles of *K. pneumoniae* isolates have revealed that apart from the minor clones they can divided into three major groups. In gel digestion of the chromosomal DNA was performed with *Xba*I. Cluster analysis was performed with Fingerprinting II Informatix (BioRad, Hercules, CA, USA). Sequenced isolates are indicated with asterisks (*).

**Figure 2 antibiotics-12-00479-f002:**
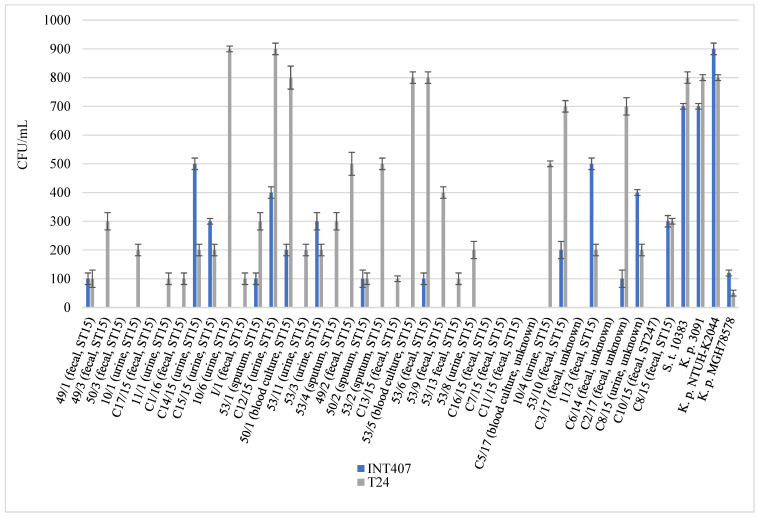
The extent of cell internalization of 39 carbapenem-resistant *K. pneumoniae* isolates in the INT407 epithelial and in the T24 bladder carcinoma cell lines, after a 3 h co-incubation. *S. typhimurium* 10383 (*S. t.* 10383), *K. pneumoniae* 3091 (*K. p.* 3091) and *K. pneumoniae* strain NTUH-K2044 (*K. p.* NTUH-K2044) were used as a positive control and *K. pneumoniae* strain MGH78578 (*K. p.* MGH78578) was used as a negative control.

**Figure 3 antibiotics-12-00479-f003:**
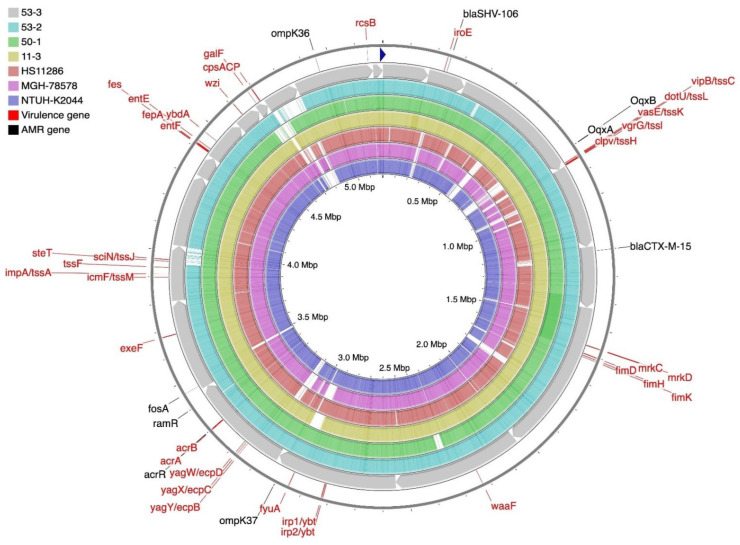
Genomic comparison map of the *K. pneumoniae* strain 53/3, strain 53/2, strain 50/1 and strain 11/3, with relative localization of the predicted virulence determinant genes. From the outside, the dark grey shows the predicted antimicrobial resistance (AMR) genes (black label), based on ResFinder results. Red labels show the predicted virulence genes, based on VFDB analysis. The second circle (grey) shows the genomic map of the *K. pneumoniae* strain 53/3 chromosome. The third circle (light blue) represents the genomic map of the *K. pneumoniae* strain 53/2 chromosome. The fourth circle (light green) shows the genomic map of the *K. pneumoniae* strain 50/1 chromosome, while the fifth circle (yellow) shows the genomic map of the *K. pneumoniae* strain 11/3 chromosome. The three innermost circles represent the chromosomes of the reference strains, like *K. pneumoniae* strain SH11286 (red), *K. pneumoniae* strain MGH78578 (pink) and *K. pneumoniae* strain NTUH-K2044 (purple). Circular genomic maps of the investigated *K. pneumoniae* isolates were obtained using the CGView Server.

**Table 1 antibiotics-12-00479-t001:** Occurrence of 13 virulence-associated genes in the genomes of the investigated *Klebsiella pneumoniae* clinical isolates (n = 39).

*K. pneumoniae* Isolates	PCR Amplification												
	*fimH-1*	*mrkD*	*mrkA*	*mrk J*	*cf29a*	*allS*	*entB*	*iutA*	*traT*	*rmpA*	*uge*	*magA*	*wziK24*
53/1	+	+	+	+	-	+	+	+	+	-	+	-	+
53/2	+	+	+	+	-	+	+	-	-	-	+	-	+
53/3	+	+	+	+	-	+	+	-	+	+	+	+	+
53/4	+	-	-	-	-	+	+	-	+	-	+	-	+
53/5	+	+	+	+	-	+	+	-	+	-	+	-	+
53/6	+	+	+	+	-	+	+	-	-	-	+	-	+
53/8	+	+	+	+	-	+	+	-	-	-	+	-	+
53/9	+	+	+	+	-	+	+	-	-	-	+	-	+
53/10	+	+	+	+	-	+	+	-	-	-	+	-	+
53/11	+	+	+	+	-	+	+	-	+	+	+	+	+
53/13	+	+	+	+	-	+	+	-	-	-	+	-	+
50/1	+	+	+	+	-	+	+	-	-	-	+	-	+
50/2	+	+	+	+	-	+	+	-	-	-	+	-	+
50/3	+	+	+	+	-	+	+	-	+	+	+	+	+
I/1	+	+	+	+	-	+	+	-	-	-	+	-	+
49/1	+	+	+	+	-	+	+	-	-	-	+	-	+
49/2	+	-	-	-	-	+	-	-	+	-	+	-	+
49/3	+	+	+	+	-	+	+	-	-	-	+	-	+
10/1	+	+	+	+	-	+	+	-	-	-	+	-	+
10/4	-	-	-	-	-	+	+	-	-	-	+	-	+
10/6	+	+	+	+	-	+	+	-	-	-	+	-	+
11/1	+	+	+	+	-	+	+	-	+	+	+	+	+
11/3	+	+	+	+	-	+	+	-	-	-	+	-	+
C6/14	+	+	+	+	-	+	+	-	+	-	+	-	+
C7/15	+	+	+	+	+	+	+	-	+	-	+	-	+
C8/15	-	+	+	+	-	+	+	+	-	-	+	-	+
C10/15	+	+	+	+	-	+	+	-	-	-	+	-	+
C11/15	+	+	+	+	+	+	+	-	+	-	+	-	+
C12/15	+	+	+	+	-	+	+	-	+	-	+	-	+
C13/15	+	+	+	+	+	+	+	-	+	-	+	-	+
C14/15	+	+	+	+	-	+	+	-	-	-	+	-	+
C15/15	+	+	+	+	-	+	+	-	+	-	+	-	+
C16/15	+	+	+	+	+	+	+	-	+	-	+	-	+
C17/15	+	+	+	+	-	+	+	-	+	-	+	-	+
C18/15	+	+	+	+	-	+	+	-	-	-	+	-	+
C1/16	+	+	+	+	-	+	-	-	-	-	+	-	+
C2/17	+	+	+	+	-	+	+	-	+	-	+	-	+
C3/17	+	+	+	+	-	+	+	-	+	-	+	-	+
C5/17	+	+	+	+	-	+	+	-	-	-	+	-	+
NTUH-K2044	+	+	+	+	+	+	+	+	+	+	+	+	-
MGH 78578	+	+	+	+	+	+	-	-	-	-	+	-	-

**Table 2 antibiotics-12-00479-t002:** Prevalence of virulence factors according to clinical *K. pneumoniae* isolates (n = 39).

*K. pneumoniae* Isolates	Type 1 Fimbriae	Type 3 Fimbriae	Biofilm Production	Siderophores Production		Serum Resistance	HMV
				Enterobactin	Aerobactin		
53/1	+	+	high	+	+	+	-
53/2	+	+	high	+	-	-	-
53/3	+	+	high	+	-	+	+
53/4	+	-	poor	+	-	+	-
53/5	+	+	high	+	-	+	-
53/6	+	+	high	+	-	-	-
53/8	+	+	high	+	-	-	-
53/9	+	+	high	+	-	-	-
53/10	+	+	high	+	-	-	-
53/11	+	+	high	+	-	+	+
53/13	+	+	high	+	-	-	-
50/1	+	+	high	+	-	-	-
50/2	+	+	high	+	-	-	-
50/3	+	+	high	+	-	+	+
I/1	+	+	high	+	-	-	-
49/1	+	+	high	+	-	-	-
49/2	+	-	poor	-	-	+	-
49/3	+	+	high	+	-	-	-
10/1	+	+	high	+	-	-	-
10/4	-	-	poor	+	-	-	-
10/6	+	+	high	+	-	-	-
11/1	+	+	high	+	-	+	+
11/3	+	+	high	+	-	-	-
C6/14	+	+	high	+	-	+	-
C7/15	+	+	high	+	-	+	-
C8/15	-	+	medium	+	+	-	-
C10/15	+	+	high	+	-	-	-
C11/15	+	+	high	+	-	+	-
C12/15	+	+	high	+	-	+	-
C13/15	+	+	high	+	-	+	-
C14/15	+	+	high	+	-	-	-
C15/15	+	+	high	+	-	+	-
C16/15	+	+	high	+	-	+	-
C17/15	+	+	high	+	-	+	-
C18/15	+	+	high	+	-	-	-
C1/16	+	+	high	-	-	-	-
C2/17	+	+	high	+	-	+	-
C3/17	+	+	medium	+	-	+	-
C5/17	+	+	high	+	-	-	-
NTUH-K2044	+	+	high	+	+	+	+
MGH 78578	+	+	high	-	-	-	-

**Table 3 antibiotics-12-00479-t003:** Comparative genomic analysis of *K. pneumoniae* strain 11/3, strain 50/1, strain 53/2 and strain 53/2.

	*K. pneumoniae* Strain 11/3 (Faecal)	*K. pneumoniae* Strain 50/1 (Blood Culture)	*K. pneumoniae* Strain 53/2 (Sputum)	*K. pneumoniae* Strain 53/3 (Urine)
Genome length (bp)	5,517,254	5,379,211	5,269,114	5,201,283
No. Contigs	41	18	18	19
Average contig length (bp)	134,567	298,845	292,728	273,751
Average GC content (%)	57.3	57.4	57.4	57.4
Genes	5180	5022	4908	4844
tRNA	81	73	81	63
CDS	5098	4948	4826	4780
Plasmid types	IncFIB(K), IncFII(K), ColpVC	IncFII(K)	ColpVC	IncFII(K)
Mutation found	*acrR*, *ramR*, o*mpK35*, *ompK36*	*acrR*, *ramR*, o*mpK35*, *ompK36*	*acrR*, o*mpK35*, *ompK36*	*acrR*, *ramR*, o*mpK35*, *ompK36*
Antimicrobial resistance genotypes	*ompK35*, *ompK36*, *bla*_CTX-M-15_, *bla*_SHV-106_, *bla*_VIM-4_, *ramR*, *acrR*	*ompK35*, *ompK36*, *bla*_CTX-M-1*5*_, *bla*_SHV-106_, *ramR*, *acrR*	*ompK35*, *ompK36*, *bla*_CTX-M-15_, *bla*_SHV-106_, *bla*_VIM-4_, *acrR*	*ompK35*, *ompK36*, *bla*_CTX-M-15_, *bla*_SHV-106_, *fosA*, *oqxA*, *oqxB*, *ramR*, *acrR*
Special virulence genes	*manB*, *manC*, *wbbM*	*mrkB*, *mrkH*, *mrkJ*	–	*cpsAPC*, *galF*, *irp1/ybt*, *irp2/ybt*, *wzi*
Accession number	JAJTNS000000000	JAJTNT000000000	JAJTNR000000000	JACTNU010000000

**Table 4 antibiotics-12-00479-t004:** Origin and Sequence Type (ST) of the 39 clinical *K. pneumoniae* isolates involved in this study.

Strain No.	Year of Isolation	Isolated from	ST
10/1	2010	urine	ST 15
10/4	2010	urine	ST 15
10/6	2010	urine	ST 15
11/1	2010	urine	ST 15
11/3	2010	faecal	ST 15
53/1	2011	sputum	ST 15
53/2	2011	sputum	ST 15
53/3	2011	urine	ST 15
53/4	2011	sputum	ST 15
53/5	2011	blood culture	ST 15
53/6	2011	faecal	ST 15
53/8	2011	urine	ST 15
53/9	2011	faecal	ST 15
53/10	2011	faecal	ST 15
53/11	2011	urine	ST 15
53/13	2011	faecal	ST 15
50/1	2011	blood culture	ST 15
50/2	2011	sputum	ST 15
50/3	2011	faecal	ST 15
I/1	2011	faecal	ST 15
49/1	2011	faecal	ST 15
49/2	2011	faecal	ST 15
49/3	2011	faecal	ST 15
C6/14	2014	faecal	ST 15
C7/15	2015	faecal	ST 15
C8/15	2015	faecal	ST 15
C10/15	2015	faecal	ST 15
C11/15	2015	faecal	ST 15
C12/15	2015	urine	ST 15
C13/15	2015	faecal	ST 15
C14/15	2015	urine	ST 15
C15/15	2015	urine	ST 15
C16/15	2015	faecal	ST 15
C17/15	2015	faecal	ST 15
C18/15	2015	urine	ST 15
C1/16	2016	faecal	ST 15
C2/17	2017	faecal	ST 15
C3/17	2017	faecal	ST 15
C5/17	2017	blood culture	ST 15

## Data Availability

All data associated with this study are presented in the main text or Supporting information.

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
