# Peer review of "Virulence Characteristics and Molecular Typing of Carbapenem-Resistant ST15 Klebsiella pneumoniae Clinical Isolates, Possessing the K24 Capsular Type"

_antibiotics, 2023, doi:10.3390/antibiotics12030479_

Round 1

Reviewer 1 Report

This work is well established as a precise and inquisitive strategy that yields a large amount of data on the resistance phenomena associated with carbapenem-resistant Klebsiella pneumoniae strains. In general the work is acceptable, however, there are some considerations that must be taken into account.

In the work, an analysis of the internalization process of Klebsiella cells in human tissue was carried out, with the purpose of determining the growth physiology in a context of interaction with the host cells, later indicated in the materials and methods section that under these experimental conditions, two types of antibiotics were used (a b-lactam and an aminoglysoside). Considering that the variety of resistance mechanisms that Klebsiella pneumoniae has demonstrated in multiple works, it is necessary to justify the reason for the use of these antibiotics and why not another class such as quinolones or sulfonamides or another class of antibiotics with another target in the physiology of the bacteria . Additionally, the work indicates that the strains used were resistant to other classes of antibiotics.

Table 3 specifies the mutations found in the strains where a comparative genomic analysis was performed. How is it explained that the mutations occurred in the same region? Are they exactly the same mutations or are they circumscribed to the same region? These issues should be clarified in the discussion section. Could it be a region more prone to accumulate mutations under a context of selection pressure with the presence of antimicrobials? This issue should be addressed in the discussion section.

The document worked with the sequencing of the genome of several strains that were subsequently subjected to analysis, for which several ResFinder 4.1 and Pathogen Wach software were used, additionally it is indicated that "Nucleotide sequences for validated Klebsiella genus virulence genes were downloaded from the Virulence Factor Database (VFDB)”. In reference to this analysis, it is pertinent to indicate that it is a robust comparative analysis based on the information present in databases, however, it is necessary to consider that the use of these tools biases the analysis towards sequences and fragments associated with virulence and/or resistance. A global analysis of the sequence by alignment makes it possible to identify other mutations in other parts of the genome that, although they cannot be directly associated as the cause of a virulent or resistant phenotype, it is necessary to consider the interaction phenomena with other genes and, therefore, if they would come into the potential category of mutations in cryptic resistance genes. It is therefore necessary to add a text to the discussion that clarifies that in future research the bioinformatics analysis strategy must consider the entire genome to broaden the analysis of mutations to potentially cryptic genes associated with virulence and/or resistance phenotypes.

Author Response

Detailed Responses to Editor and Reviewers

Manuscript ID: Antibiotics - 2224731

Response to the Editor’s comments

We are very pleased to resubmit for publication the revised version of Virulence characteristics and molecular typing of carbapenem-resistant ST15 Klebsiella pneumoniae clinical isolates, possessing the K24 capsular type” (Antibiotics - 2224731) Marianna Horváth, Tamás Kovács, József Kun, Attila Gyenesei, Ivelina Damjanova, Zoltán Tigyi and György Schneider to be considered for publication as an original article in MDPI Antibiotics. We carefully considered your comments hoping our revision has improved the paper to a level of your satisfaction.

Again, we appreciate the opportunity to revise our work for consideration for publication in MDPI Antibiotics.

The authors thank the reviewers for their rapid and constructive reviews of the manuscript. Here are our detailed responses to the reviewer's issues. Reviewers comments are reported in red.

Response to Reviewer #1 Comments

This work is well established as a precise and inquisitive strategy that yields a large amount of data on the resistance phenomena associated with carbapenem-resistant Klebsiella pneumoniae strains. In general the work is acceptable, however, there are some considerations that must be taken into account.

Response: Authors are grateful for the positive comments and supportive opinion.

In the work, an analysis of the internalization process of Klebsiella cells in human tissue was carried out, with the purpose of determining the growth physiology in a context of interaction with the host cells, later indicated in the materials and methods section that under these experimental conditions, two types of antibiotics were used (a b-lactam and an aminoglysoside). Considering that the variety of resistance mechanisms that Klebsiella pneumoniae has demonstrated in multiple works, it is necessary to justify the reason for the use of these antibiotics and why not another class such as quinolones or sulfonamides or another class of antibiotics with another target in the physiology of the bacteria. Additionally, the work indicates that the strains used were resistant to other classes of antibiotics.

Response: Thank you for your legitimate comment. During the cell internalization assay we have used penicillin, streptomycin and neomycin for cultivation the cell lines. For the internalization tests media without antibiotics were used first. Generally, gentamicin is used to kill non-internalised bacterium cells in the medium. If the tested bacterium is resistant to gentamicin than polymixin B can be one choice of drug as it was earlier published. Advantage of both antibiotics is that they can not penetrate inside the eukaryotic cell.

Table 3 specifies the mutations found in the strains where a comparative genomic analysis was performed. How is it explained that the mutations occurred in the same region? Are they exactly the same mutations or are they circumscribed to the same region? These issues should be clarified in the discussion section. Could it be a region more prone to accumulate mutations under a context of selection pressure with the presence of antimicrobials? This issue should be addressed in the discussion section.

Response: The described differences among the isolates with common clonal origins could be a result of an evolutionary process happening in the community and subsequently brought into the hospital. Occurrence of the same mutations in the affected resistance genes can represent it. Therefore based on your suggestion we inserted the following paragraph in to Discussion as a the third paragraph of this section:   

“Common clonal origin of the K. pneumoniae isolates were supported by the fact that all isolates (n=39) belonged to ST15 and possessed the K24 capsular type and their antibiograms showed high similarities to each other (Supplementary Table 2a and b). Furthermore, identified pointmutations (location inside the gene is labelled with: p.XY) in the resistance genes of the four sequenced isolates 11/3, 50/1, 53/2 and 53/3 were localized on to the same positions affecting the carbapenem resistance in the ompK35 (p.A217S) and ompK36 (p.I128M, p.I70M) [46]. This was also the case in other two inactive resistance genes, namely the acrR and ramR. acrR gene (1492 - 4449 bp) is conferred resistance to fluoroquinolones [47, 48] located on a plasmid, and was affected with the same seven pointmutations (p.F197I, p.K201M, p.L195V, p.G164A, p.R173G, p.F172S and p.P161R). Only one mutation (p.A19V) was identified in all sequenced isolates in ramR gene, that is conferred to tigecycline resistance [47, 48, 49].”

Just for an example in case of the isolate 11/3, that we have identifed with ResFinder 4.1:

The document worked with the sequencing of the genome of several strains that were subsequently subjected to analysis, for which several ResFinder 4.1 and Pathogen Wach software were used, additionally it is indicated that "Nucleotide sequences for validated Klebsiella genus virulence genes were downloaded from the Virulence Factor Database (VFDB)”. In reference to this analysis, it is pertinent to indicate that it is a robust comparative analysis based on the information present in databases, however, it is necessary to consider that the use of these tools biases the analysis towards sequences and fragments associated with virulence and/or resistance. A global analysis of the sequence by alignment makes it possible to identify other mutations in other parts of the genome that, although they cannot be directly associated as the cause of a virulent or resistant phenotype, it is necessary to consider the interaction phenomena with other genes and, therefore, if they would come into the potential category of mutations in cryptic resistance genes. It is therefore necessary to add a text to the discussion that clarifies that in future research the bioinformatics analysis strategy must consider the entire genome to broaden the analysis of mutations to potentially cryptic genes associated with virulence and/or resistance phenotypes.

Response: Thank you for your helpful comment.

We hope that these revisions improve the paper such that you and the reviewers now deem it worth of publication in MDPI Antibiotics.

Reviewer 2 Report

Dear Authors

Thank you for your manuscript submission. Your study is well-designed and well-presented.

I just want to suggest two effective papers which can be useful for your manuscript. It is recommended to read and add the following papers to References section of the manuscript to have fruitful Introduction an Discussion sections:

The prevalence of type 3 fimbriae in Uropathogenic Escherichia coli isolated from clinical urine samples. Meta Gene. 2021 Jun 1;28:100881.

Virulence factors, antibiotic resistance patterns, and molecular types of clinical isolates of Klebsiella Pneumoniae. Expert Rev Anti Infect Ther. 2022 Mar;20(3):463-472. doi: 10.1080/14787210.2022.1990040. Epub 2021 Oct 28. PMID: 34612762.

Author Response

Detailed Responses to Editor and Reviewers

Manuscript ID: Antibiotics - 2224731

Response to the Editor’s comments

We are very pleased to resubmit for publication the revised version of Virulence characteristics and molecular typing of carbapenem-resistant ST15 Klebsiella pneumoniae clinical isolates, possessing the K24 capsular type” (Antibiotics - 2224731) Marianna Horváth, Tamás Kovács, József Kun, Attila Gyenesei, Ivelina Damjanova, Zoltán Tigyi and György Schneider to be considered for publication as an original article in MDPI Antibiotics. We carefully considered your comments hoping our revision has improved the paper to a level of your satisfaction.

Again, we appreciate the opportunity to revise our work for consideration for publication in MDPI Antibiotics.

The authors thank the reviewers for their rapid and constructive reviews of the manuscript. Here are our detailed responses to the reviewer's issues. Reviewers comments are reported in red.

Response to Reviewer #2 Comments

Dear Authors

Thank you for your manuscript submission. Your study is well-designed and well-presented.

I just want to suggest two effective papers which can be useful for your manuscript. It is recommended to read and add the following papers to References section of the manuscript to have fruitful Introduction an Discussion sections:

The prevalence of type 3 fimbriae in Uropathogenic Escherichia coli isolated from clinical urine samples. Meta Gene. 2021 Jun 1;28:100881.

Virulence factors, antibiotic resistance patterns, and molecular types of clinical isolates of Klebsiella Pneumoniae. Expert Rev Anti Infect Ther. 2022 Mar;20(3):463-472. doi: 10.1080/14787210.2022.1990040. Epub 2021 Oct 28. PMID: 34612762.

Response: Thank you for your remark, we reformulated this section and we inserted a new references (Khonsari et al., 2021 – Line 265; Ahmadi et al., 2022 – Line 268).

We hope that these revisions improve the paper such that you and the reviewers now deem it worth of publication in MDPI Antibiotics.

Reviewer 3 Report

First of all I really would like to appreciate the authors of the manuscript for such a wonderful work and its representation.

I would like to suggest some minor comments before the manuscript being accepted for publication. The comments are as follows:

1) Line 125: Full form of CIM should be mentioned 

2) Figure 1: The quality of the figure should be improved 

3) In the section of " presence of virulence associated genes i.e. 2.3 line 136 to 153 the results should either be represented in number or in percentage in the sentences as you are already mention in the tabulated form so I feel only representation in percentage will be fine.

4) Similarly you might do the changes in the section 2.4

5) In the 2.5 the things are getting repeated, if you have already tabulated everything in the table 3 than why you are repeating in the form of sentence?

6) In material and method section, the first section of identification of the organism was mentioned but if you are not showing the results than what's the importance of showing it into the material and method section?

7) In section 4.4 try to mention the sequence of the primer used.

Try to recheck the manuscript for any grammatical and spelling mistake.

Author Response

Detailed Responses to Editor and Reviewers

Manuscript ID: Antibiotics - 2224731

Response to the Editor’s comments

We are very pleased to resubmit for publication the revised version of Virulence characteristics and molecular typing of carbapenem-resistant ST15 Klebsiella pneumoniae clinical isolates, possessing the K24 capsular type” (Antibiotics - 2224731) Marianna Horváth, Tamás Kovács, József Kun, Attila Gyenesei, Ivelina Damjanova, Zoltán Tigyi and György Schneider to be considered for publication as an original article in MDPI Antibiotics. We carefully considered your comments hoping our revision has improved the paper to a level of your satisfaction.

Again, we appreciate the opportunity to revise our work for consideration for publication in MDPI Antibiotics.

The authors thank the reviewers for their rapid and constructive reviews of the manuscript. Here are our detailed responses to the reviewer's issues. Reviewers comments are reported in red.

Response to Reviewer #3 Comments

First of all I really would like to appreciate the authors of the manuscript for such a wonderful work and its representation.

I would like to suggest some minor comments before the manuscript being accepted for publication. The comments are as follows:

1) Line 125: Full form of CIM should be mentioned 

Response: Thank you for your helpful comment. We have corrected it (Line 115).

2) Figure 1: The quality of the figure should be improved 

Response: Thank you for your helpful comment. We have modified Figure 1 accordingly (Page5).

3) In the section of " presence of virulence associated genes i.e. 2.3 line 136 to 153 the results should either be represented in number or in percentage in the sentences as you are already mention in the tabulated form so I feel only representation in percentage will be fine.

Response: Thank you for your remark, we deleted it. For better understanding and traceability we intended to summarize this data both as a text and in a table form.

4) Similarly you might do the changes in the section 2.4

Response: Thank you for your remark, we deleted it.

5) In the 2.5 the things are getting repeated, if you have already tabulated everything in the table 3 than why you are repeating in the form of sentence?

Response: For better understanding and traceability we intended to summarize this data both as a text and in a table form.

6) In material and method section, the first section of identification of the organism was mentioned but if you are not showing the results than what's the importance of showing it into the material and method section?

Response: Thank you for your helpful comment. This disinformation from our side was corrected in the following way: Isolates were analysed by MALDI-TOF MS and the results were compared with standard conventional identification. All of the isolates (n=39) were identified at species level [log (score value) ≥ 2.0]. The results of standard biochemical tests showed that, all of the isolates were negative for indole probe, methyl red test, ornithine- and arginine- decarboxylase test and motility test. All of the isolates were positive for adonite test, Voges-Proskauer test, citrate test, malonate test, urease test, lysine-decarboxylase test and saccharose and lactose test (Lines 104 – 109).

7) In section 4.4 try to mention the sequence of the primer used.

Response: Thank you for your remark, we added the sequence of the MLST primers (Line 373 – 379) rpoB F: Vic3: GGCGAAATGGCWGAGAACCA, R: Vic2: GAGTCTTCGAAGTTGTAACC; gapA F: gapA173: TGAAATATGACTCCACTCACGG, R: gapA181: CTTCAGAAGCGGCTTTGATGGCTT; mdh F: mdh130: CCCAACTCGCTTCAGGTTCAG, R: mdh867: CCGTTTTTCCCCAGCAGCAG; pgi F: pgi1F: GAGAAAAACCTGCCTGTACTGCTGGC, R: pgi1R: CGCGCCACGCTTTATAGCGGTTAAT; phoE F: phoE604.1: , ACCTACCGCAACACCGACTTCTTCGG, R: phoE604.2 TGATCAGAACTGGTAGGTGAT; infB F: infB1F: CTCGCTGCTGGACTATATTCG, R: infB1R: CGCTTTCAGCTCAAGAACTTC and tonB F: tonB1F: CTTTATACCTCGGTACATCAGGTT, R: tonB2R: ATTCGCCGGCTGRGCRGAGAG.

We hope that these revisions improve the paper such that you and the reviewers now deem it worth of publication in MDPI Antibiotics.
